# Genome-Wide Association Study Identifies Candidate Loci Associated with Opioid Analgesic Requirements in the Treatment of Cancer Pain

**DOI:** 10.3390/cancers14194692

**Published:** 2022-09-27

**Authors:** Daisuke Nishizawa, Takeshi Terui, Kunihiko Ishitani, Shinya Kasai, Junko Hasegawa, Kyoko Nakayama, Yuko Ebata, Kazutaka Ikeda

**Affiliations:** 1Addictive Substance Project, Tokyo Metropolitan Institute of Medical Science, Tokyo 156-8506, Japan; 2Division of Internal Medicine, Department of Medicine, Higashi-Sapporo Hospital, Sapporo 003-8585, Japan

**Keywords:** opioids, analgesics, single-nucleotide polymorphisms, genome-wide association study, cancer pain

## Abstract

**Simple Summary:**

Pain is experienced by ~55% of patients who undergo anti-cancer treatment. However, many genetic factors that are involved in the efficacy of opioid analgesics in the treatment of cancer pain remain unidentified. The aim of our genome-wide association study (GWAS) was to comprehensively explore genetic variations that are associated with opioid analgesic requirements in the treatment of cancer pain. We identified several single-nucleotide polymorphisms (SNPs) that are associated with opioid analgesic requirements, two of the most potent of which were the rs1283671 and rs1283720 SNPs in the *ANGPT1* gene region, and SNPs in the *SLC2A14* gene were also associated with the phenotype. These results indicate that these SNPs in the *ANGPT1* and *SLC2A14* genes could serve as markers that predict the efficacy of opioid analgesics in the treatment of cancer pain.

**Abstract:**

Considerable individual differences have been widely observed in the sensitivity to opioids. We conducted a genome-wide association study (GWAS) in patients with cancer pain to identify potential candidate single-nucleotide polymorphisms (SNPs) that contribute to individual differences in opioid analgesic requirements in pain treatment by utilizing whole-genome genotyping arrays with more than 650,000 markers. The subjects in the GWAS were 428 patients who provided written informed consent and underwent treatment for pain with opioid analgesics in a palliative care unit at Higashi-Sapporo Hospital. The GWAS showed two intronic SNPs, rs1283671 and rs1283720, in the *ANGPT1* gene that encodes a secreted glycoprotein that belongs to the angiopoietin family. These two SNPs were strongly associated with average daily opioid requirements for the treatment of pain in both the additive and recessive models (*p* < 5.0000 × 10^−8^). Several other SNPs were also significantly associated with the phenotype. In the gene-based analysis, the association was significant for the *SLC2A14* gene in the additive model. These results indicate that these SNPs could serve as markers that predict the efficacy of opioid analgesics in cancer pain treatment. Our findings may provide valuable information for achieving satisfactory pain control and open new avenues for personalized pain treatment.

## 1. Introduction

Pain is defined by the International Association for the Study of Pain as “An unpleasant sensory and emotional experience associated with, or resembling that associated with, actual or potential tissue damage” [1]. Although pain is reportedly experienced by 55% of patients who undergo anti-cancer treatment and by 66% of patients who have advanced, metastatic, or terminal disease, many more cancer patients may potentially experience pain according to a recent systematic review and meta-analysis [2]. The World Health Organization recommends that any opioid may be considered for the maintenance of pain relief (alone or combined with nonsteroidal anti-inflammatory drugs [NSAIDs] and/or paracetamol) in adults (including older persons) and adolescents with cancer-related pain to achieve sustained, effective, and safe pain control, depending on clinical assessment and pain severity [3]. However, their effects are not uniform across all patients. Patient responses to opioid medications vary by patient, specific medication, and other factors [3,4,5]. Such significant variations in opioid responsiveness or sensitivity may influence analgesic effects that are necessary to achieve sufficient pain relief, which may hinder the clinical management of pain. Although twin studies by Angst et al. indicated that genetic effects on the opioid-induced elevation of heat or cold-pressor pain thresholds accounted for 12–60% of the observed response variance [6,7], many genetic factors remain to be elucidated.

To identify genetic variations, mostly single-nucleotide polymorphisms (SNPs), that are associated with human opioid sensitivity or responsiveness and related phenotypes, many candidate gene association studies have been conducted [8,9,10]. The most extensively investigated genetic variation is the functional A118G (rs1799971) SNP in the µ-opioid receptor gene, *OPRM1*. A recent meta-analysis indicated that G-allele carriers (AG+GG) of this SNP required higher opioid doses for cancer pain management than AA homozygotes [11]. Genetic polymorphisms in the *COMT* gene, which encodes catechol-*O*-methyltransferase, have also been often investigated, and SNPs in this gene were found to be associated with responsiveness to opioids even in advanced cancer patients [12,13]. Moreover, several other SNPs in the *ABCB1*, *IL6*, *GCH1*, and *TAOK3* gene have also been found to be associated with the response to opioids in patients with cancer pain using a candidate gene approach [14].

Genome-wide association studies (GWASs) focus on genetic variations in all regions of the human genome without regard to preexisting theories about the relationship between variations and the phenotypes of interest [15]. GWASs can also be used to investigate genetic factors that are related to individual differences in analgesic effects of opioids. However, only a few studies have examined human opioid responsiveness and sensitivity based on GWASs. One of these studies investigated associations between SNPs and total morphine requirements as a quantitative trait locus in a retrospective population of opioid-naive children (4–18 years old) who had undergone day surgery tonsillectomy and adenoidectomy. This study identified an association between the rs795484 and rs1277441 SNPs at the *TAOK3* gene locus and morphine dose [16]. Additionally, we performed a GWAS of traits that are linked to opioid sensitivity. We chose participants who were scheduled to undergo aesthetic orthognathic surgery for mandibular prognathism [17] and found that the best candidate was the rs2952768 SNP, located near the *METTL21A* (*FAM119A*) and *CREB1* gene regions [18]. A GWAS was also conducted in patients with cancer from 11 European countries using a prospective cross-sectional multinational, multicenter design [19]. The rs12948783 SNP, which corresponds to chromosome 17 upstream of the *RHBDF2* gene, had the strongest association with responsiveness to opioids in this study’s patients who received opioid treatment for moderate or severe pain [20]. However, no GWAS has yet been conducted in populations other than of European origin, such as Asian populations.

In the present study, we conducted a GWAS in patients with cancer pain to identify potential genetic variants that contribute to the efficacy of opioid analgesics that are used to treat pain, reflected by analgesic requirements.

## 2. Materials and Methods

### 2.1. Patients

#### 2.1.1. Patients Who Underwent Cancer Pain Treatment with Opioid Analgesics

We enrolled 428 adult patients (20–94 years old, 213 males and 215 females) who suffered from various types of cancer and were hospitalized at Higashi-Sapporo Hospital (Hokkaido, Japan) for the treatment of cancer pain in 2017–2019 and were apparently Japanese. Higashi-Sapporo Hospital was established in 1983 as a hospital that specializes in cancer care, particularly palliative care. Characteristics of the hospital and palliative cancer care were detailed in previous reports [21,22]. All of the patients who were recruited in the present study were treated with opioid analgesics, and many were also appropriately treated with NSAIDs and/or other supplementary analgesics for the treatment of pain. We excluded patients who were considered to be unsuitable by their primary care doctors. Detailed demographic and clinical data of the subjects are provided in Table 1. Peripheral blood samples were collected from these subjects for gene analysis.

The study was conducted according to guidelines of the Declaration of Helsinki and approved by the Institutional Review Board or Ethics Committee of Higashi-Sapporo Hospital and Tokyo Metropolitan Institute of Medical Science (Tokyo, Japan). Written informed consent was obtained from all of the patients.

#### 2.1.2. Patient Characteristics and Clinical Data

In the patient subjects, we obtained data on surgical history, treatment history, pain status (e.g., presence/absence of somatic pain, visceral pain, and neuropathic pain), drug treatments, and disease status (e.g., lung cancer, breast cancer, stomach cancer, etc.; Table 1). Some of the patients were affected by multiple diseases.

The treatment of pain was mainly conducted by administering opioid analgesics (e.g., morphine and oxycodone; Table 1). Various types of drugs, such as NSAIDs (e.g., loxoprofen and diclofenac) and/or other supplementary analgesics (e.g., pregabalin and dexamethasone), were also administered at the discretion of primary care doctors if required. To allow intersubject comparisons of opioid analgesic doses that were required for cancer pain treatment, the opioid doses were converted to equivalent doses of oral morphine as described in Appendix A, based on previous reports with slight modification [23,24,25]. The total dose of converted opioid analgesics that were administered was calculated daily, and the total dose of analgesics was calculated as a daily average based on the amount of 5 days of administration, 3–7 days before blood collection. This average total dose was used as the endpoint of opioid requirements for the genetic association analysis in the present study. Doses of analgesics that were administered were normalized to body weight. The detailed clinical data of the subjects are detailed in Table 1.

### 2.2. Whole-Genome Genotyping, Quality Control, and Gene-Based and Gene-Set Analyses

#### 2.2.1. Whole-Genome Genotyping and Quality Control

A total of 428 DNA samples from the patients were used for genotyping. Total genomic DNA was extracted from whole-blood samples using standard procedures. The extracted DNA was dissolved in TE buffer (10 mM Tris-HCl and 1 mM ethylenediaminetetraacetic acid [EDTA], pH 8.0). The DNA concentration was adjusted to 50 ng/μL for whole-genome genotyping using a NanoDrop ND-1000 Spectrophotometer (NanoDrop Technologies, Wilmington, DE, USA).

According to the manufacturer’s recommendations, whole-genome genotyping was performed by utilizing the Infinium Assay II with an iScan system (Illumina, San Diego, CA, USA). Infinium Asian Screening Array-24 v. 1.0 BeadChips (one kind) were utilized to genotype 428 patient samples (total markers: 659,184). Numerous copy number variation markers were included in the BeadChips, but the majority of the BeadChips were for SNP markers on the human autosome or sex chromosome.

GenomeStudio with the Genotyping v. 2.0.4 module (Illumina) was used to examine data for samples that had their entire genomes genotyped to assess quality of the findings. Following data cleaning, samples with genotype call rates less than 0.95 were not included in the remaining studies. No patient samples were consequently disregarded for subsequent investigations (i.e., all of the 428 samples met the 0.95 quality control cutoff, and the entire samples were considered for further association analyses). In the subsequent association analyses, markers with genotype call frequencies less than 0.95 and “Cluster sep” (a measure of genotype cluster separation) values less than 0.1 were not included. The patient samples retained a total of 650,504 SNP markers after this screening step. Markers were further filtered based on a test of Hardy–Weinberg equilibrium (HWE). Markers with *p* values (df = 1) less than 0.001 were considered to be deviated in the HWE tests and thus were excluded. Finally, the analysis only included the 648,817 SNPs that passed the full filtration procedure.

A log quantile-quantile (QQ) *p*-value plot as a result of the GWAS for the entire sample was subsequently drawn to check the pattern of the generated *p*-value distribution in the association studies between the SNPs and phenotype, in which the observed *p* values against the values that were expected from the null hypothesis of a uniform distribution, calculated as –log10 (*p* value), were plotted for each model. All of the plots were mostly concordant with the expected line (y = x), especially over the range of 0 < –log10 (*p* value) < 2 for each model, indicating no apparent population stratification of the samples that were used in the study (Appendix A). Additionally, results of the GWAS are presented as Manhattan plots, in which the –log10 (*p* value) for each SNP on each chromosome was plotted simultaneously.

#### 2.2.2. Gene-Based and Gene-Set Analyses

Gene-based and gene-set approaches were adopted with Multi-marker Analysis of GenoMic Annotation (MAGMA) v. 1.06 [26], which is also available on the Functional Mapping and Annotation of Genome-Wide Association Studies (FUMA GWAS) v. 1.3.9 platform [27], to better understand genetic backgrounds and molecular mechanisms that underlie complex traits, such as opioid requirements in cancer pain treatment. To examine the combined relationship between all genetic markers in the gene and the phenotype, genetic marker data were aggregated to the level of full genes in the gene-based analysis. Similarly, individual genes were compiled into groupings of genes with similar biological, functional, or other properties for the gene-set analysis. Gene-set analysis can thus shed light on the role that particular biological pathways or cellular processes may play in the genetic basis of a trait [26]. In these analyses, associations were explored for genes on autosomes 1–22 and the X chromosome, and the window of the genes to assign SNPs was set to 20 kb, thereby assigning SNPs within the 20 kb window of the gene (both sides) to that gene. For the reference panel, the 1000 Genome Phase3 EAS population was selected (http://ftp.1000genomes.ebi.ac.uk/vol1/ftp/release/20130502/ accessed on 26 July 2022). In the gene-set analysis, gene sets were defined using the Molecular Signatures Database (MSigDB) v. 7.5.1 (https://www.gsea-msigdb.org/gsea/msigdb accessed on 26 September 2022) [28]. A total of 10,678 gene sets (curated gene sets: 4761, GO terms: 5917) from MsigDB were tested. In both analyses, Bonferroni correction for multiple testing was performed for all tested genes and gene sets. Adjusted values of *p* < 0.05 in the results were considered significant.

### 2.3. Statistical Analysis

In the association studies of patients who underwent cancer pain treatment with opioids, the total dose of analgesics as a daily average that was calculated based on the amount of 5 days of administration, 3–7 days before blood collection, was used as an index of opioid sensitivity. Before the analyses, the quantitative values of total analgesic requirements that were equivalent to oral morphine (mg/kg) were natural log-transformed for approximation to the normal distribution according to the following formula: *Value for analyses = Ln (1 + total analgesic requirements on average per day [mg/kg])*. To explore associations between the SNPs and phenotypes, linear regression analyses were conducted, in which total analgesic requirements as a daily average (mg/kg; log-transformed) and genotype data for each SNP were incorporated as dependent and independent variables, respectively. Additionally, weight and binary data on the presence/absence of neuropathic pain were incorporated as covariables because these parameters were strongly associated with the dependent variable in a preliminary study (data not shown). Additive, dominant, and recessive genetic models were used for the analyses because of previously insufficient knowledge about genetic factors that are associated with opioid sensitivity. Male genotypes were not included in the analysis of X chromosome markers, whereas both male and female individuals were included in the association study for autosomal markers. PLINK v. 1.07 (https://zzz.bwh.harvard.edu/plink/index.shtml accessed on 28 July 2022) [29], gPLINK v. 2.050 [29], and Haploview v. 4.2 [30] were used to perform the statistical analyses. The criterion for significance in the GWAS was set at *p* < 5 × 10^−8^, which is widely known to be a conventional criterion for the level of significance in GWASs [31,32]. Additionally, Hardy–Weinberg equilibrium was tested using Exact Tests for genotypic distributions of SNPs that were significantly associated with the phenotype.

### 2.4. Additional in Silico Analysis

#### 2.4.1. Power Analysis

G*Power 3.0.5 software (https://www.psychologie.hhu.de/arbeitsgruppen/allgemeine-psychologie-und-arbeitspsychologie/gpower accessed on 26 September 2022) was used to perform preliminary statistical power analyses [33]. When the type I error probability was set at 1 × 10^−7^ (less than 5 × 10^−8^) and the sample size was 428, power assessments for the linear regression analysis showed that the expected power (1 minus type II error probability) was over 99.4% for a Cohen’s conventional “middle” effect size of 0.15 [34]. When Cohen’s conventional “small” effect size was 0.02, however, for the same type I error probability and sample sizes of 428, the estimated power decreased to 0.7%. In contrast, assuming the same type I error probability and sample sizes of 428 to obtain 80% power, the predicted effect size was 0.0919. For effect sizes from large to moderately small but not too small, a single analysis in the preceding study was anticipated to detect true associations with the phenotype with 80% statistical power, although the precise effect size has not been well understood in cases of SNPs that significantly influence opioid sensitivity.

#### 2.4.2. Linkage Disequilibrium Analysis

Linkage disequilibrium (LD) analysis was performed using Haploview v. 4.2 [30] for a total of 428 samples from patients who underwent cancer pain treatment with opioids for the locus of the ~243 kbp region that was annotated as the *ANGPT1* gene and its flanking region on chromosome 8 based on an annotation file that was provided by Illumina (San Diego, CA, USA) to identify relationships between SNPs in the *ANGPT1* gene region. The commonly used *D*′ and *r*^2^ values were pairwise calculated using the genotype dataset for each SNP to estimate the strength of LD between SNPs. Linkage disequilibrium blocks were defined as in a previous study [35].

#### 2.4.3. Reference of Databases

Several databases and bioinformatic tools were referenced to more thoroughly examine the candidate SNP that may be related to human opioid analgesic sensitivity, including the National Center for Biotechnology Information (NCBI) database (http://www.ncbi.nlm.nih.gov accessed on 28 July 2022), Genotype-Tissue Expression (GTEx) portal (https://www.gtexportal.org/home/; accessed on 28 July 2022) [36], HaploReg v. 4.1 (https://pubs.broadinstitute.org/mammals/haploreg/haploreg.php; accessed on 28 July 2022) [37], and SNPinfo Web Server (https://snpinfo.niehs.nih.gov/; accessed on 28 July 2022) [38]. The GTEx project, an ongoing effort to create a comprehensive public resource to research tissue-specific gene expression and regulation [36], is the basis for the GTEx portal, which offers open access to such data as gene expression, quantitative trait loci, and histology images. HaploReg is a tool for investigating non-coding genomic annotations at variations in haplotype blocks, such as potential regulatory SNPs at disease-associated sites [37]. The SNPinfo Web Server is a set of web-based SNP selection tools (freely available at https://snpinfo.niehs.nih.gov/; accessed on 28 July 2022) where investigators can specify genes or linkage regions and select SNPs based on GWAS results, LD, and predicted functional characteristics of both coding and non-coding SNPs [38].

## 3. Results

### 3.1. Identification of Genetic Polymorphisms Associated with Opioid Analgesic Requirements in the Treatment of Cancer Pain by GWAS

We comprehensively explored genetic variations that were associated with opioid analgesic requirements per day in the treatment of cancer pain in a total of 428 patients. A total of 648,817 SNPs that met the quality control standards in the GWAS of all patients were examined for relationships with the phenotype in the additive, dominant, and recessive models. The top 20 potential SNPs for the additive, dominant, and recessive models are shown in Table 2, Table 3 and Table 4, respectively. Significant associations were found for the rs1283671 and rs1283720 SNPs on chromosome 8 and the GSA-rs72752701 (rs72752701) SNP on chromosome 9 in the additive model (*p* = 3.876 × 10^−11^ for rs1283671, *p* = 4.007 × 10^−11^ for rs1283720, *p* = 3.879 × 10^−8^ for GSA-rs72752701; Table 2, Figure 1a). Significant associations were also found for the kgp11947181 (rs79524268) SNP on chromosome 6, 9:133013640 (rs2417163) SNP on chromosome 9, and rs763315292 SNP on chromosome 10 in the dominant model (*p* = 1.592 × 10^−8^ for kgp11947181, *p* = 1.730 × 10^−8^ for 9:133013640, *p* = 3.807 × 10^−8^ for rs763315292; Table 3, Figure 1b). Significant associations were also found in the recessive model for the same three SNPs as in the additive model (*p* = 7.698 × 10^−11^ for rs1283671, *p* = 7.698 × 10^−11^ for rs1283720, *p* = 3.814 × 10^−8^ for GSA-rs72752701; Table 4, Figure 1c). Many of the examined SNPs’ computed -log10 *p* values (observed *p* values), which are based on the null hypothesis of a uniform distribution in the QQ plot, differed from the predicted values (Appendix A). The values for SNPs with significant relationships in Table 2, Table 3 and Table 4 (rs1283671 and rs1283720) and other SNPs were obviously higher than predicted values (Appendix A). The gene that encodes angiopoietin-1, *ANGPT1*, was in the vicinity of the rs1283671 and rs1283720 SNPs (i.e., the best candidate SNPs in the additive and recessive models). Meanwhile, the kgp11947181 SNP, the best candidate SNP in the dominant model, was located in an intergenic region on chromosome 6 (Appendix A). The rs1283671 and rs1283720 SNPs were in strong LD with each other (*r*^2^ = 0.97), although these two SNPs were not in strong LD (*r*^2^ < 0.80) with other neighboring SNPs within and around the *ANGPT1* gene (Appendix A). As shown in Table 2 and Table 4, an increase in the minor allele carry in the rs1283671 and rs1283720 SNPs was associated with greater opioid analgesic requirements in cancer pain treatment.

### 3.2. Identification of Genes and Gene Sets Associated with Opioid Analgesic Requirements in the Treatment of Cancer Pain by Gene-Based and Gene-Set Analyses

Considering that the effects of individual markers tend to be too weak to be detected by comprehensive analyses, such as GWASs, that target only single polymorphisms, we conducted gene-based and gene-set analyses, which are statistical methods that are used to analyze multiple genetic markers simultaneously to determine their joint effect. In both analyses, using MAGMA software [26], which was made accessible in the FUMA GWAS platform [27], we investigated genes and gene sets that were related to the amount of opioid analgesics that were required daily for the management of cancer pain in a total of 428 individuals. As a result, 648,817 SNPs from the selected candidate genes and gene sets in the additive, dominant, and recessive models were included in the analyses of all patients. The top 20 candidate genes that were found in each genetic model by the gene-based analysis are listed in Table 5. In the additive model, *SLC2A14*, the top candidate gene, was significantly associated with the phenotype (adjusted *p* = 0.03722; Table 5, Figure 2a). However, in both the dominant and recessive models, none of the genes were significantly associated with the phenotype (Table 5, Figure 2b,c), although the association with the *SLC2A14* gene was marginally significant in the recessive model (*p* = 0.07384; Table 5). The top 10 candidate gene sets that were found in each genetic model using the gene-set analysis are listed in Appendix A. However, in none of the genetic models were any of the gene sets significantly related to the phenotype (Appendix A).

## 4. Discussion

To identify potential genetic variants that contribute to the requirements of opioid analgesics that were used to treat cancer pain, we conducted a GWAS in patient subjects who were diagnosed with various types of cancers (Table 1). The GWAS results suggested that carriers of minor alleles of the rs1283671 and rs1283720 SNPs within the *ANGPT1* gene region were less sensitive to opioid analgesics and thus required greater amounts of daily opioid analgesics in cancer pain treatment (Table 2 and Table 4). Our GWAS also identified other potent SNPs that are possibly associated with opioid sensitivity, including GSA-rs72752701 (rs72752701) in the additive and recessive models (Table 2 and Table 4) and kgp11947181 (rs79524268), 9:133013640 (rs2417163), and rs763315292 in the dominant model (Table 3). Additionally, numerous SNPs in or near the *SLC2A14* gene region collectively increased the amount of analgesic that was required for cancer pain management (Table 5). No relevant gene sets that may have been related to the trait were identified (Appendix A). These findings show that in the genetic etiology of the trait, individual genes did not properly aggregate to groups of genes that shared specific biological, functional, or other features. Future studies with larger sample sizes are required to find candidate gene sets that are associated with the phenotype and corroborate the significant results that were identified in our GWAS of single SNPs and gene-set analysis.

Previous studies identified several genetic variations that are associated with phenotypes that are related to opioid responsiveness in the treatment of cancer pain [11,12,13,14,20]. Included in those candidate genetic variations are the rs1799971, rs1319339, and rs34427887 SNPs in the *OPRM1* gene, rs4680, rs4818, rs4633, rs4646316, and rs35478083 SNPs in the *COMT* gene, rs1045642 SNP in the *ABCB1* gene, −174G>C (rs1800795) SNP in the *IL6* gene, rs8007267, rs3783641, and rs10483639 SNPs in the *GCH1* gene, rs12948783 SNP in the *RHBDF2* gene, and rs1277441 and rs795484 SNPs in the *TAOK3* gene [11,12,13,14,20]. Among these SNPs, rs1799971, rs4680, rs4633, rs4646316, rs1045642, rs1800795, rs8007267, rs3783641, rs10483639, rs12948783, and rs795484 SNPs were included in the SNP array that was used in the present study. However, no SNPs except rs8007267 (*p* = 0.007411) and rs3783641 (*p* = 0.006942) in the *GCH1* gene were even nominally significantly associated with opioid analgesic requirements (*p* > 0.05) in our association analysis results (details not shown). The *GCH1* gene encodes GTP cyclohydrolase 1, which is the first and rate-limiting enzyme in tetrahydrobiopterin (BH4) biosynthesis. Although these two SNPs, with the other rs10483639 SNP, were reported to be associated with opioid therapy initiation in cancer pain patients [39], these three SNPs were not shown to be associated with the responsiveness to opioids itself. These results might indicate the general difficulty of replicating results of human genetic association studies, probably because of the heterogeneity of study designs, diagnoses of different types of cancer, pain severity, types of opioid analgesics used, and genetic background of the subjects, among other factors.

The best candidate SNPs in the present study were rs1283671 and rs1283720, which are found in the intronic region of the *ANGPT1* gene on chromosome 8. These SNPs had the lowest *p* values among the candidate SNPs. Angiopoietin-1, a secreted glycoprotein that is a member of the angiopoietin family, is encoded by the *ANGPT1* gene. Angiopoietin-1 is an angiogenic growth factor with antipermeability and anti-inflammatory properties [40]. Chronic or uncontrolled inflammation plays an important role in many diseases [41], including stroke [42]. When the angiopoetin-1 receptor, Tie2, becomes inactivated, important molecular brakes are released in the endothelium, which in turn potentiate inflammation and vascular leakage [43]. *ANGPT1* mRNA is known to be broadly expressed in fat, the lungs, the prostate, and several other tissues in humans, according to the NCBI database. Mice that were engineered to lack angiopoietin-1 exhibited angiogenic deficits [44]. A missense mutation (807G>T, A119S) that was identified in a family was associated with hereditary angioedema in humans [45]. Meanwhile, angiogenesis may be protected against by angiotensin-converting enzyme (ACE) inhibitors [46], and the inhibition of ACE activity might prevent or even reverse hypoglycemia-associated autonomic failure (HAAF) [47]. β-endorphin has also been reported to influence the autonomic response to hypoglycemia via opioid receptor activation [48]. The opioid receptor antagonist naloxone improved counterregulatory responses [49] and prevented HAAF [48] in humans. Therefore, both the inhibition of ACE activity and antagonism of opioid receptors may protect against angiogenesis and HAAF in humans, suggesting that angiogenesis, in which angiopoietin-1 is involved, could also be modulated by actions of opioids. Therefore, a possible explanation for the present results may be that genetic variations in the *ANGPT1* gene could influence individual differences in angiogenic effects, which in turn may lead to individual differences in opioid analgesic efficacy. However, to date, no genetic variations of the *ANGPT1* gene have been reported to be related to functions of the opioid system. The rs1283671 and rs1283720 SNPs (i.e., the best candidate SNPs in the present study) have not been previously reported to be associated with any phenotypes to date. Although the rs1283671 and rs1283720 SNPs were in strong LD with each other (*r^2^* = 0.97), these two SNPs were not in strong LD (*r^2^* < 0.80) with other neighboring SNPs in our data (Appendix A). When these SNPs were referenced in HaploReg v. 4.1 [37] (accessed on 2 August 2022), they were in strong LD (*r^2^* ≥ 0.80) with 20 neighboring SNPs (data not shown). HaploReg v. 4.1 also showed that the rs1283671 and rs1283720 SNPs could change eight and three motifs, respectively, for DNA-binding proteins. Nevertheless, none of these SNPs were significantly associated with mRNA expression levels of any genes in any tissues, according to the GTEx portal (accessed on 2 August 2022), suggesting that these SNPs unlikely influence variations in opioid sensitivity among individuals by influencing the mRNA expression of some genes.

Our GWAS identified several other potent SNPs beyond rs1283671 and rs1283720 that were significantly associated with opioid requirements in cancer pain patients, including GSA-rs72752701 (rs72752701) in the additive and recessive models (Table 2 and Table 4) and kgp11947181 (rs79524268), 9:133013640 (rs2417163), and rs763315292 in the dominant model (Table 3). Among these, the GSA-rs72752701, kgp11947181, and 9:133013640 SNPs are located in intergenic regions, whereas rs763315292 is located in the SEC23 interacting protein gene (*SEC23IP*) region. Although no significant expression quantitative trait loci (eQTLs) were found for kgp11947181 and rs763315292, the GSA-rs72752701 SNP is associated with mRNA expression of the heat shock protein family A (Hsp70) member 5 (*HSPA5*) gene, and the 9:133013640 SNP is associated with mRNA expression of the hemicentin 2 (*HMCN2*) and long intergenic non-protein coding RNA 963 (*LINC00963*) genes, according to the GTEx portal [36]. The *SEC23IP*, *HMCN2*, and *LINC00963* genes have not been previously reported to be related to the opioid system or its receptors. In a previous quantitative mass spectrometry proteomics study of purified synaptosomes that were isolated from the caudate in two groups of rhesus macaques that were chronically infected with simian immunodeficiency virus (SIV) and differed with regard to morphine treatment regimens, the upregulation of HSPA5 was found in the SIV+morphine group, indicating an increase in cellular stress as a result of the SIV/morphine interaction, thereby leading to central nervous system dysfunction [50]. This previous study suggests that the expression and function of HSPA5 could interact with the opioid system, and alterations of *HSPA5* mRNA expression through the GSA-rs72752701 SNP might influence the function of opioids.

Although none of the top 30 candidate SNPs in the additive model in the present study included SNPs that were annotated as this gene (Table 2), the *SLC2A14* gene was significantly associated with the trait in the gene-based analysis of all patients (Table 5, Figure 2a). The *SLC2A14* gene encodes solute carrier family 2 member 14 (SLC2A14), which is a member of the glucose transporter (GLUT) family. Members of the GLUT family, including SLC2A14, are highly conserved integral membrane proteins that transport hexoses, such as glucose and fructose, into all mammalian cells. However, the functional relationship between this protein and the opioid system is unknown. *SLC2A14* mRNA is known to be highly expressed in the testis [51], followed by bone marrow, the urinary bladder, the placenta, the gall bladder, the appendix, and the brain, according to the NCBI database. In human genetic studies, SNPs in the *SLC2A14* gene region were reported to be associated with some diseases, including Alzheimer’s disease and inflammatory bowel disease [52,53]. Notably, several SNPs in the *SLC2A14* gene region were also reported to be associated with alcohol dependence at the *p* ≤ 10^−4^ level of significance [54], which is known to be at least partly pertinent to the opioid system in its etiology [55]. Since alcohol dependence could be caused by the rewarding effects of alcohol, in which the opioid system is also involved, an antagonist of opioid receptors, naltrexone, is widely used for the treatment of alcohol dependence [56]. One of the associated SNPs, rs11612319, was also included in our GWAS and not strongly but rather modestly associated with the phenotype (*p* = 4.325 × 10^−4^). Although the impact of this SNP on SLC2A14 function and the opioid system remains unknown, this SNP may have some influence on both the risk for alcohol dependence and opioid sensitivity.

One limitation of the present study was the relatively small sample size, which might have led to an inability to detect significant associations in the gene-set analysis. Other limitations include heterogeneity of the primary disease and the level of cancer pain. In the present study, patients with various types of cancer were recruited, and a GWAS was conducted in all samples. With much larger sample sizes, GWASs could be performed in a particular subset of patients, such as only in patients with lung cancer (Table 1), which was actually difficult in the present study because only a small number of such patients (<100) would have been included in the GWAS. The lack of data on the level of cancer pain in each patient in the present study also hampered the ability to conduct a GWAS only in patients with cancer pain of relatively homogeneous severity. Notwithstanding these limitations, the present study identified several SNPs and the *SLC2A14* gene that were significantly associated with the phenotype. Regardless of the type of cancer or severity of cancer pain, these SNPs and this gene might commonly impact opioid sensitivity, and they were associated with the need for opioids to treat cancer pain in analyses of whole samples.

The results of the present study suggest that the rs1283671 and rs1283720 SNPs of the *ANGPT1* gene could be clinical markers that predict analgesic requirements, in which the minor alleles of these SNPs are possibly associated with lower opioid sensitivity and thus greater requirements for opioid analgesics in the treatment of cancer pain. Before clinically applying these findings, further studies should be designed to better predict analgesic requirements, such as studies of the construction and validation of prediction formulas for analgesic requirements, similar to previous studies [57,58], by considering the influence of other possible candidate SNPs and non-genetic factors.

## 5. Conclusions

In conclusion, our GWASs discovered SNPs and a gene that were associated with opioid analgesic requirements in the treatment of cancer pain, including the *ANGPT1* rs1283671 and rs1283720 SNPs and *SLC2A14* gene. Although the present results need to be corroborated by more research with larger sample sizes, these findings indicate that these SNPs in the *ANGPT1* and *SLC2A14* genes could serve as markers that predict the efficacy of opioid analgesics in the treatment of cancer pain.

## Figures and Tables

**Figure 1 cancers-14-04692-f001:**
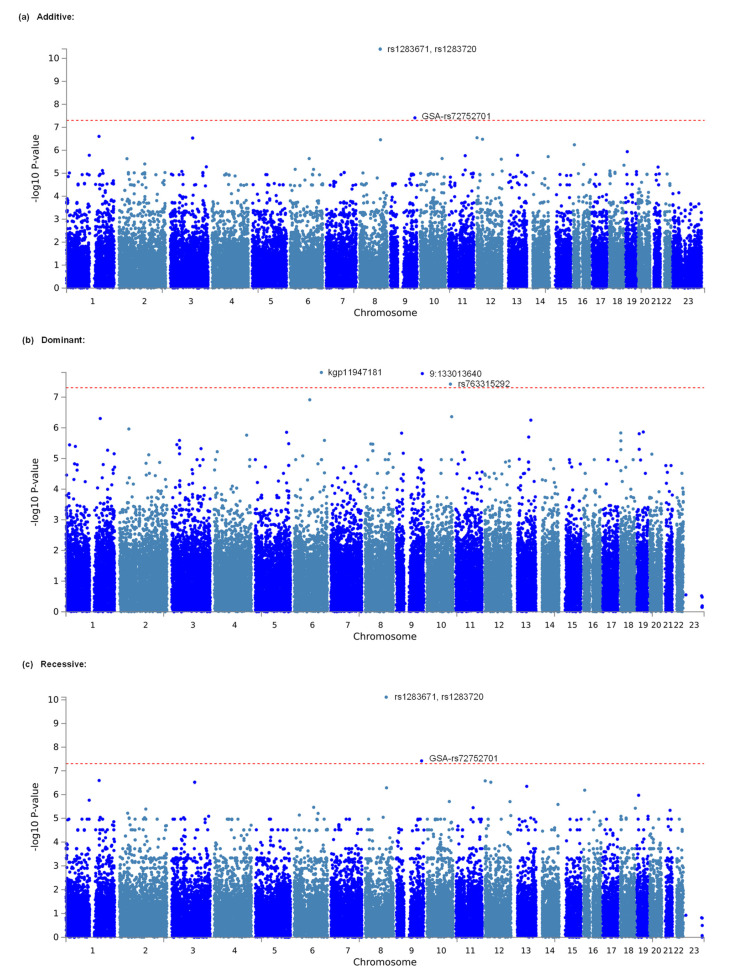
Manhattan plot of GWAS outcomes. (**a**) Analytical plot in the additive model. (**b**) Analytical plot in the dominant model. (**c**) Analytical plot in the recessive model. The dashed red line shows the significant association threshold.

**Figure 2 cancers-14-04692-f002:**
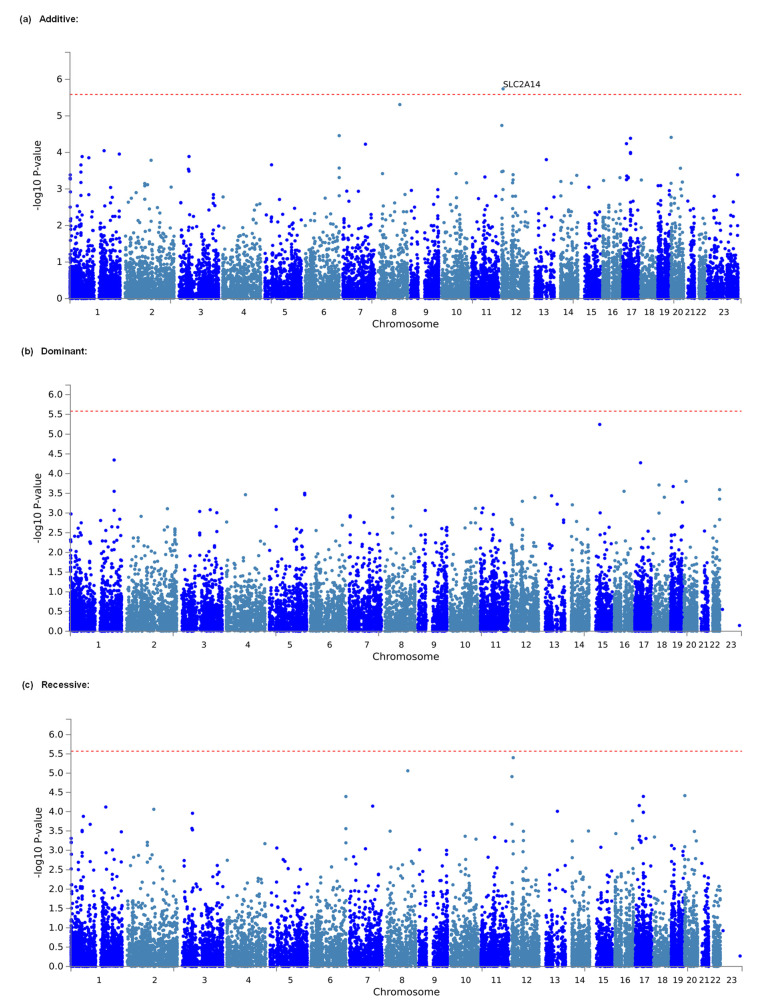
Manhattan plot of results of gene-based analysis. (**a**) Analytical plot in the additive model. (**b**) Analytical plot in the dominant model. (**c**) Analytical plot in the recessive model. The dashed red line shows the significant association threshold.

**Table 1 cancers-14-04692-t001:** Demographic and clinical data of patient subjects.

Demographic Data:	*n*	Minimum	Maximum	Mean	SD	Median		
Gender								
male	213							
female	215							
**Age [years]**	428	20	94	71.59	12.33	72.50		
**Height [cm]**	423	136	181	157.61	8.56	158.00		
**Weight [kg]**	421	30	110	50.56	11.07	50.00		
**Diagnosis (primary disease):**	** *n* **				**Diagnosis (primary disease):**			* **n** *
lung cancer	81				breast cancer			60
stomach cancer	26				pancreas cancer			25
prostate cancer	25				colon cancer			22
bladder cancer	17				rectal cancer			13
hepatocellular cancer	12				esophageal cancer			10
uterus cancer	9				ovary cancer			8
others	120							
**Status and side effects:**	**absence**	**presence**						
somatic pain	38	390						
visceral pain	346	82						
neuropathic pain	327	101						
nausea	193	138						
constipation	113	245						
**Breakdown of opioids:**	** *n* **	**morphine**	**oxicodone**	**fentanyl**	**tapentadol**	**tramadol**	**methadone**	**hydromorphone**
Number of administered patients	426	97	209	107	10	31	5	41
**Administration of opioids:**	** *n* **	**Minimum**	**Maximum**	**Mean**	**SD**	**Median**		
Total dose of opioids [mg/kg]	419	0.060241	32.407407	1.74	3.09	0.8163265		
(converted to oral morphine)								
**Administration of other drugs:**	**absence**	**presence**						
NSAIDs	137	291						
supplementary analgesics	171	257						

**Table 2 cancers-14-04692-t002:** Top 30 candidate SNPs selected from the GWAS (additive model).

Model	Rank	CHR	SNP	Position	*p*	Related Gene	Genotype (Patients)			Phenotype (Mean)		
							A/A	A/B	B/B	A/A	A/B	B/B
Additive	1	8	rs1283671	107421459	3.876 × 10^−11^ *	*ANGPT1*	5	87	327	2.375	0.873	0.703
Additive	2	8	rs1283720	107460100	4.007 × 10^−11^ *	*ANGPT1*	5	89	325	2.375	0.86	0.705
Additive	3	9	GSA−rs72752701	125992274	3.879 × 10^−8^ *		2	50	367	3.011	0.776	0.743
Additive	4	1	rs3917744	169608752	2.485 × 10^−7^	*SELP*	2	51	366	2.832	0.778	0.744
Additive	5	12	rs11061996	1836333	2.846 × 10^−7^	*CACNA2D4,LRTM2*	2	47	370	2.774	0.714	0.752
Additive	6	3	rs61355450	117954317	0.000000294		2	36	381	2.869	0.786	0.744
Additive	6	3	rs6798512	118019819	0.000000294		2	36	381	2.869	0.786	0.744
Additive	8	12	GSA-rs117577524	30549638	3.316 × 10^−7^		2	27	390	2.869	0.794	0.744
Additive	9	8	rs4620259	108979439	0.000000349		5	74	340	2.175	0.828	0.722
Additive	10	16	rs74007038	6785856	5.776 × 10^−7^	*RBFOX1*	5	79	335	2.076	0.781	0.733
Additive	11	19	rs75017760	8940941	0.00000115	*MUC16*	2	64	353	2.883	0.708	0.755
Additive	12	13	rs9529111	66652839	0.000001648	*PCDH9*	19	131	269	1.426	0.672	0.752
Additive	13	1	1:118813000	118270377	0.000001653		2	39	378	2.917	0.805	0.741
Additive	14	11	rs10501687	88799011	0.000001725	*GRM5*	22	159	237	1.307	0.768	0.7
Additive	15	14	kgp19644675	100077596	0.000001896	*EVL*	5	57	357	1.927	0.851	0.727
Additive	16	10	rs10749151	114017465	0.000002292		7	105	307	1.833	0.704	0.752
Additive	17	6	rs17059990	99980179	0.0000023	*MCHR2*	2	65	352	2.817	0.866	0.726
Additive	18	2	2:42166453	41939313	0.00000233	*C2orf91*	35	162	220	1.168	0.74	0.702
Additive	19	12	rs1554080	128183013	0.00000245		5	70	344	1.814	0.701	0.754
Additive	20	2	rs11692586	134159434	0.000003951		11	102	306	1.593	0.751	0.73
Additive	21	16	rs75384045	55308868	0.000004132		3	52	364	2.242	0.85	0.732
Additive	22	18	rs117231913	79691289	0.000004475	*CTDP1*	3	55	361	2.421	0.693	0.754
Additive	23	3	rs73184492	189405146	0.000005259		1	53	364	3.509	0.939	0.718
Additive	24	21	rs2834573	34668083	0.000005392		2	93	324	2.646	0.696	0.764
Additive	25	6	kgp17238235	27175279	0.000006765		4	85	330	1.919	0.795	0.734
Additive	26	6	kgp3351958	122754906	0.00000681		2	36	380	2.838	0.709	0.751
Additive	27	11	rs10831496	88824823	0.000007262	*GRM5*	24	167	228	1.251	0.757	0.707
Additive	28	1	rs75731751	187443485	0.000007626		1	50	368	3.509	0.944	0.725
Additive	29	18	rs989644	4231203	0.000007922	*DLGAP1*	1	49	368	3.509	0.784	0.742
Additive	30	12	12:132932218	132355632	0.000008031		1	20	395	3.509	0.78	0.743

Model, the genetic model in which the GWAS selected candidate SNPs; CHR, chromosome number; Related gene, the gene that is closest to the SNP location; A/A, each SNP’s minor allele is homozygous; A/B, each SNP’s major allele is heterozygous; B/B, each SNP’s major allele is homozygous; * *p* < 5 × 10^−8^.

**Table 3 cancers-14-04692-t003:** Top 30 candidate SNPs selected from the GWAS (dominant model).

Model	Rank	CHR	SNP	Position	*p*	Related Gene	Genotype (Patients)			Phenotype (Mean)		
							A/A	A/B	B/B	A/A	A/B	B/B
Dominant	1	6	kgp11947181	139582822	1.592 × 10^−8^ *		0	2	417	NA	3.176	0.746
Dominant	2	9	9:133013640	130251361	1.730 × 10^−8^ *		0	2	417	NA	3.118	0.746
Dominant	3	10	rs763315292	119912126	3.807 × 10^−8^ *	*SEC23IP*	0	3	416	NA	2.62	0.744
Dominant	4	6	6:80456944	79747227	1.223 × 10^−7^		0	4	415	NA	2.434	0.742
Dominant	5	7	7:118263373	118623319	2.066 × 10^−7^		0	2	417	NA	2.953	0.747
Dominant	6	10	rs77717582	126393947	4.341 × 10^−7^		0	23	396	NA	1.411	0.72
Dominant	7	1	rs755431252	175365091	4.979 × 10^−7^	*TNR*	0	3	416	NA	2.677	0.744
Dominant	8	13	13:88770953	88118698	5.634 × 10^−7^		0	4	415	NA	2.109	0.745
Dominant	9	2	rs63751260	47403309	0.000001091	*MSH2*	0	2	416	NA	2.883	0.749
Dominant	10	19	rs146815072	32830639	0.000001368	*SLC7A9*	0	4	415	NA	2.289	0.743
Dominant	11	5	rs145778277	161311253	0.0000014	*GABRB2*	0	13	406	NA	1.516	0.734
Dominant	12	18	18:3452222	3452225	0.000001471	*TGIF1*	0	2	415	NA	2.859	0.75
Dominant	13	9	GSA-rs10967750	27166161	0.000001495	*TEK*	0	5	414	NA	1.993	0.743
Dominant	14	19	19:12311461	12200646	0.000001562	*LOC100289333*	1	9	408	2.471	1.537	0.738
Dominant	15	4	JHU_4.166326986	165405835	0.000001738	*CPE*	3	54	362	0.942	1.079	0.708
Dominant	16	13	13:77841745	77267610	0.000002003	*MYCBP2*	0	4	412	NA	2.031	0.747
Dominant	17	6	rs192596782	155305908	0.000002575	*TFB1M*	0	3	413	NA	2.397	0.745
Dominant	18	3	rs117341459	39630484	0.00000258		0	4	415	NA	2.144	0.744
Dominant	19	18	rs4632226	3989188	0.000002665	*DLGAP1,DLGAP1-AS4*	1	27	390	0.179	1.287	0.722
Dominant	20	5	kgp5717569	172042597	0.0000033	*STK10*	1	31	386	2.125	1.136	0.725
Dominant	21	8	GSA-rs147608943	30052419	0.000003361		0	5	414	NA	1.955	0.743
Dominant	22	8	8:40320989	40463470	0.000003413		0	4	414	NA	2.163	0.739
Dominant	23	3	kgp2760268	26300600	0.000003538		0	3	416	NA	2.323	0.747
Dominant	24	1	kgp15630468	18045740	0.000003604		0	12	407	NA	1.479	0.737
Dominant	25	1	kgp15515856	47636211	0.000004042		0	8	411	NA	1.59	0.742
Dominant	26	3	3:40166299	40124808	0.00000439	*MYRIP*	0	2	415	NA	2.646	0.75
Dominant	27	3	3:40027538	39986047	0.000004617	*MYRIP*	0	2	417	NA	2.646	0.749
Dominant	28	3	3:149817230	150099443	0.000004786		0	4	415	NA	2.075	0.745
Dominant	29	19	19:12306462	12195647	0.000005029	*LOC100289333*	1	14	404	2.471	1.345	0.733
Dominant	29	19	19:12352578	12241763	0.000005029		1	14	404	2.471	1.345	0.733

Model, the genetic model in which the GWAS selected the candidate SNPs; CHR, chromosome number; Related gene, the gene that is closest to the SNP location; A/A, each SNP’s minor allele is homozygous; A/B, each SNP’s major allele is heterozygous; B/B, each SNP’s major allele is homozygous; NA, not available; * *p* < 5 × 10^−8^.

**Table 4 cancers-14-04692-t004:** Top 30 candidate SNPs selected from the GWAS (recessive model).

Model	Rank	CHR	SNP	Position	*p*	Related Gene	Genotype (Patients)			Phenotype (Mean)		
							A/A	A/B	B/B	A/A	A/B	B/B
Recessive	1	8	rs1283671	107421459	7.698 × 10^−11^ *	*ANGPT1*	5	87	327	2.375	0.873	0.703
Recessive	1	8	rs1283720	107460100	7.698 × 10^−11^ *	*ANGPT1*	5	89	325	2.375	0.86	0.705
Recessive	3	9	GSA-rs72752701	125992274	3.814 × 10^−8^ *		2	50	367	3.011	0.776	0.743
Recessive	4	1	rs3917744	169608752	2.568 × 10^−7^	*SELP*	2	51	366	2.832	0.778	0.744
Recessive	5	12	rs11061996	1836333	2.672 × 10^−7^	*CACNA2D4,LRTM2*	2	47	370	2.774	0.714	0.752
Recessive	6	3	rs61355450	117954317	3.039 × 10^−7^		2	36	381	2.869	0.786	0.744
Recessive	6	3	rs6798512	118019819	3.039 × 10^−7^		2	36	381	2.869	0.786	0.744
Recessive	6	12	GSA-rs117577524	30549638	3.039 × 10^−7^		2	27	390	2.869	0.794	0.744
Recessive	9	13	rs9529111	66652839	4.469 × 10^−7^	*PCDH9*	19	131	269	1.426	0.672	0.752
Recessive	10	8	rs4620259	108979439	5.146 × 10^−7^		5	74	340	2.175	0.828	0.722
Recessive	11	16	rs74007038	6785856	6.499 × 10^−7^	*RBFOX1*	5	79	335	2.076	0.781	0.733
Recessive	12	19	rs75017760	8940941	0.000001066	*MUC16*	2	64	353	2.883	0.708	0.755
Recessive	13	1	1:118813000	118270377	0.000001725		2	39	378	2.917	0.805	0.741
Recessive	14	10	rs10749151	114017465	0.000001958		7	105	307	1.833	0.704	0.752
Recessive	15	12	rs1554080	128183013	0.000001979		5	70	344	1.814	0.701	0.754
Recessive	16	14	kgp19644675	100077596	0.000002605	*EVL*	5	57	357	1.927	0.851	0.727
Recessive	17	6	rs17059990	99980179	0.000003439	*MCHR2*	2	65	352	2.817	0.866	0.726
Recessive	18	11	rs10501687	88799011	0.000003562	*GRM5*	22	159	237	1.307	0.768	0.7
Recessive	19	18	rs117231913	79691289	0.000003767	*CTDP1*	3	55	361	2.421	0.693	0.754
Recessive	20	2	rs11692586	134159434	0.00000408		11	102	306	1.593	0.751	0.73
Recessive	21	21	rs2834573	34668083	0.000004617		2	93	324	2.646	0.696	0.764
Recessive	22	16	rs75384045	55308868	0.000005399		3	52	364	2.242	0.85	0.732
Recessive	23	2	2:42166453	41939313	0.000006107	*C2orf91*	35	162	220	1.168	0.74	0.702
Recessive	24	6	kgp3351958	122754906	0.00000622		2	36	380	2.838	0.709	0.751
Recessive	25	6	kgp17238235	27175279	0.000007322		4	85	330	1.919	0.795	0.734
Recessive	26	12	12:132932218	132355632	0.00000773		1	20	395	3.509	0.78	0.743
Recessive	27	3	rs73184492	189405146	0.000008219		1	53	364	3.509	0.939	0.718
Recessive	27	18	rs989644	4231203	0.000008219	*DLGAP1*	1	49	368	3.509	0.784	0.742
Recessive	27	19	rs112363595	56660873	0.000008219		1	18	399	3.509	0.707	0.748
Recessive	30	8	rs4734909	91861390	0.000009082		13	120	286	1.467	0.656	0.768

Model, the genetic model in which the GWAS selected the candidate SNPs; CHR, chromosome number; Related gene, the gene that is closest to the SNP location; A/A, each SNP’s minor allele is homozygous; A/B, each SNP’s major allele is heterozygous; B/B, each SNP’s major allele is homozygous; * *p* < 5 × 10^−8^.

**Table 5 cancers-14-04692-t005:** Top 20 candidate genes selected from gene-based analysis.

Model	Rank	CHR	Gene Start Position	Gene Stop Position	Gene	nSNPs	Z Statistic	*p*	*P ^a^*
Additive	1	12	7945108	8063744	*SLC2A14*	30	4.6313	1.8167 × 10^−6^	0.035151328 *
Additive	2	8	108241721	108530283	*ANGPT1*	24	4.4216	4.8995 × 10^−6^	0.094800426
Additive	3	12	1909433	1965918	*LRTM2*	19	4.1283	0.000018275	0.353602975
Additive	4	6	170131718	170201680	*ERMARD*	4	3.978	0.000034747	0.672319703
Additive	5	20	1141205	1186059	*TMEM74B*	8	3.9518	0.000038784	0.750431616
Additive	6	17	39113968	39163387	*KRT40*	7	3.938	0.000041089	0.795031061
Additive	7	17	19102674	19145839	*AC106017.1*	2	3.8561	0.000057607	1
Additive	8	7	112439202	112599971	*C7orf60*	5	3.8481	0.000059519	1
Additive	9	1	169538087	169619431	*SELP*	21	3.7461	0.000089799	1
Additive	10	17	39094669	39143144	*KRT39*	5	3.7175	0.00010059	1
Additive	11	17	39129686	39170385	*KRTAP3-3*	6	3.7019	0.00010702	1
Additive	12	1	245113007	245310466	*EFCAB2*	26	3.6928	0.00011092	1
Additive	13	3	50106341	50176454	*RBM5*	4	3.6532	0.00012949	1
Additive	14	1	60338980	60412462	*CYP2J2*	2	3.6525	0.00012986	1
Additive	15	1	93625476	93764287	*CCDC18*	6	3.6329	0.00014014	1
Additive	16	13	75838808	76076250	*TBC1D4*	42	3.6025	0.00015758	1
Additive	17	2	128828774	128973251	*UGGT1*	6	3.5918	0.00016422	1
Additive	18	5	32669176	32811819	*NPR3*	19	3.5168	0.00021839	1
Additive	19	1	54477347	54539177	*TMEM59*	3	3.5143	0.00022045	1
Additive	20	6	169837307	170122159	*WDR27*	15	3.4629	0.00026718	1
Dominant	1	15	41454923	41542941	*EXD1*	5	4.3882	5.7145 × 10^−6^	0.108884083
Dominant	2	1	212983483	213040991	*C1orf227*	5	3.9154	0.000045129	0.859887966
Dominant	3	17	28236218	28455470	*EFCAB5*	10	3.876	0.000053086	1
Dominant	4	20	9498036	9839689	*PAK7*	86	3.6046	0.00015634	1
Dominant	5	18	28936740	29014875	*DSG4*	17	3.548	0.00019409	1
Dominant	6	19	12253879	12320064	*ZNF136*	13	3.5248	0.00021191	1
Dominant	7	22	51041182	51086607	*ARSA*	19	3.4757	0.00025478	1
Dominant	8	16	47091614	47197908	*NETO2*	6	3.4494	0.00028092	1
Dominant	9	1	213011597	213092705	*FLVCR1*	9	3.4493	0.00028106	1
Dominant	10	5	171268553	171453877	*FBXW11*	21	3.4178	0.00031566	1
Dominant	11	4	93198402	93245329	*RP11-9B6.1*	3	3.3954	0.00034267	1
Dominant	12	5	171449077	171635390	*STK10*	64	3.3934	0.00034512	1
Dominant	13	13	50466842	50530626	*SPRYD7*	3	3.3782	0.00036482	1
Dominant	14	8	37680786	37727422	*BRF2*	11	3.372	0.00037312	1
Dominant	15	18	54794293	54837531	*BOD1L2*	15	3.3544	0.00039763	1
Dominant	16	12	116375711	116735143	*MED13L*	33	3.349	0.00040551	1
Dominant	17	22	51019114	51072409	*MAPK8IP2*	25	3.3244	0.00044301	1
Dominant	18	12	55503465	55544586	*OR9K2*	7	3.2881	0.00050434	1
Dominant	19	19	57301445	57372096	*PEG3*	14	3.2739	0.00053034	1
Dominant	20	13	77598792	77921185	*MYCBP2*	28	3.2397	0.00059826	1
Recessive	1	12	7945108	8063744	*SLC2A14*	30	4.467	3.9655 × 10^−6^	0.073841576
Recessive	2	8	108241721	108530283	*ANGPT1*	24	4.2961	8.6923 × 10^−6^	0.161859318
Recessive	3	12	1909433	1965918	*LRTM2*	19	4.2181	0.000012317	0.229354857
Recessive	4	20	1141205	1186059	*TMEM74B*	8	3.9541	0.000038414	0.715307094
Recessive	5	17	39113968	39163387	*KRT40*	7	3.9425	0.000040318	0.750761478
Recessive	6	6	170131718	170201680	*ERMARD*	4	3.9412	0.000040543	0.754951203
Recessive	7	17	19102674	19145839	*AC106017.1*	2	3.8099	0.000069518	1
Recessive	8	7	112439202	112599971	*C7orf60*	5	3.8012	0.000071998	1
Recessive	9	1	169538087	169619431	*SELP*	21	3.7878	0.000076002	1
Recessive	10	2	128828774	128973251	*UGGT1*	6	3.7536	0.000087158	1
Recessive	11	13	75838808	76076250	*TBC1D4*	42	3.7231	0.000098378	1
Recessive	12	17	39129686	39170385	*KRTAP3-3*	6	3.7097	0.00010375	1
Recessive	13	17	39094669	39143144	*KRT39*	5	3.7068	0.00010496	1
Recessive	14	3	50106341	50176454	*RBM5*	4	3.6936	0.00011057	1
Recessive	15	1	60338980	60412462	*CYP2J2*	2	3.6468	0.00013276	1
Recessive	16	16	85703690	85804735	*C16orf74*	11	3.5785	0.00017279	1
Recessive	17	12	1881123	2048002	*CACNA2D4*	41	3.5256	0.00021126	1
Recessive	18	1	93625476	93764287	*CCDC18*	6	3.5227	0.00021358	1
Recessive	19	3	47249516	47344941	*KIF9*	3	3.4581	0.00027197	1
Recessive	20	6	169837307	170122159	*WDR27*	15	3.454	0.00027618	1

Model, the genetic model in which candidate genes were selected by analysis; CHR, chromosome number; nSNPs, the number of SNPs annotated to the gene; Z Statistic, gene-based test statistic; *p ^a^*, adjusted *p* value for multiple testing; *, significant association after conservative Bonferroni correction.

## Data Availability

Data that are presented in this study are available upon request from the corresponding author.

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
