# Peer review of "Genome-Wide Association Study Identifies Candidate Loci Associated with Opioid Analgesic Requirements in the Treatment of Cancer Pain"

_cancers, 2022, doi:10.3390/cancers14194692_

Round 1

Reviewer 1 Report

This is a study of clinical samples taken from cancer patients and analyzed for SNPs associated with opioid needs. There were several loci identified, including two from ANGPT1 gene, among others. There does not appear to be any previous report of the angiopoietin system as playing a role in opioid efficacy, thus the findings are novel. There were other potential linkages as well, which also appeared novel. There are clear weaknesses, such as the lack of correlation with specific cancer types or etc., but these were also recognized by the investigators as arising from a relatively small data set.

Noted Issues

·       Line 125: please define how doses were converted to morphine equivalents.

·       Lines 146-156. Please clarify or provide data on how many samples did not meet the 0.95 quality control cutoffs. It is not clear what the actual sequencing N was, it seems like all samples passed, but it was not very clear.

·      The simple summary claims “Pain is experienced by ~55% of patients who undergo anti-cancer treatment.” It is surprisingly low and does not discuss the variance seen in metastatic and other patients. Indeed an argument could likely be made that 100% of cancer patients experience pain. Unless this is a well-accepted number, please consider removing due to ambiguity and lack of context.

·      Table 1, bottom, “drugs” is misspelled

·       Line 208: it was claimed: “The criterion for significance in the GWAS was set at p < 5 × 10-8, which is widely known to be a conventional criterion for the level of significance in GWASs.” Please provide a reference.

Author Response

Comments and Suggestions for Authors

This is a study of clinical samples taken from cancer patients and analyzed for SNPs associated with opioid needs. There were several loci identified, including two from ANGPT1 gene, among others. There does not appear to be any previous report of the angiopoietin system as playing a role in opioid efficacy, thus the findings are novel. There were other potential linkages as well, which also appeared novel. There are clear weaknesses, such as the lack of correlation with specific cancer types or etc., but these were also recognized by the investigators as arising from a relatively small data set.

Noted Issues

1) · Line 125: please define how doses were converted to morphine equivalents.

Response: According to the reviewer’s suggestion, we included an explanation of how doses were converted to morphine equivalents in the manuscript and new Supplementary Table S1. [Lines 127-129]

2) · Lines 146-156. Please clarify or provide data on how many samples did not meet the 0.95 quality control cutoffs. It is not clear what the actual sequencing N was, it seems like all samples passed, but it was not very clear.

Response: According to the reviewer’s suggestion, we clarified the data on how many samples met the 0.95 quality control cutoffs and revised the manuscript to make this clearer. [Lines 153-154]

3) · The simple summary claims “Pain is experienced by ~55% of patients who undergo anti-cancer treatment.” It is surprisingly low and does not discuss the variance seen in metastatic and other patients. Indeed an argument could likely be made that 100% of cancer patients experience pain. Unless this is a well-accepted number, please consider removing due to ambiguity and lack of context.

Response: Although the “WHO guidelines for the pharmacological and radiotherapeutic management of cancer pain in adults and adolescents” [3] report that pain is experienced by ~55% of patients who undergo anti-cancer treatment, and this seems to be a generally well-accepted number, many more patients (perhaps nearly 100% patients) could actually experience some level of pain. Therefore, we revised the text according to the reviewer’s suggestion. [Lines 45-49]

4) · Table 1, bottom, “drugs” is misspelled

Response: According to the reviewer’s suggestion, we corrected the spelling of “drugs.” [Table 1]

5) · Line 208: it was claimed: “The criterion for significance in the GWAS was set at p < 5 × 10-8, which is widely known to be a conventional criterion for the level of significance in GWASs.” Please provide a reference.

Response: According to the reviewer’s suggestion, we provided references for this sentence. [Line 214]

Reviewer 2 Report

Dear Authors,

Thank you for this interesting study.

It is my pleasure to give a green light for this publication. 

Just few questions/suggestions from my side:

- line 111 (and abstract) you claim that all patients provided their writen consent. These patients were recruited 2017-2019 and the ethical committee approval was 2018. Would you comment on that?

- I was counting on a bit deeper discussion on ANGPT1 function and possible explanation of these results. 

- Do we know anything on how these genetic variants influence the function of the receptor?

- Please add two sentences more on how these results can be used in clinical practice or what other studies should be planned before we can think of any clinical application of these data.

Regards

Author Response

Comments and Suggestions for Authors

Dear Authors,

Thank you for this interesting study.

It is my pleasure to give a green light for this publication.

Just few questions/suggestions from my side:

1) - line 111 (and abstract) you claim that all patients provided their writen consent. These patients were recruited 2017-2019 and the ethical committee approval was 2018. Would you comment on that?

Response: The date of ethical committee approval in the manuscript indicated the date of approval for the revised protocol at Higashi-Sapporo Hospital. The date of approval for the initial protocol was November 20, 2016. To avoid confusion, we revised the manuscript to mention the date of approval for the initial protocol at both Higashi-Sapporo Hospital and the Tokyo Metropolitan Institute of Medical Science. [Lines 508-510]

2) - I was counting on a bit deeper discussion on ANGPT1 function and possible explanation of these results.

Response: According to the reviewer’s suggestion, we added a deeper discussion of ANGPT1 function and possible explanations of the present results.

“Meanwhile, angiogenesis may be protected against by angiotensin-converting enzyme (ACE) inhibitors [46], and the inhibition of ACE activity might prevent or even reverse hypoglycemia-associated autonomic failure (HAAF) [47]. β-endorphin has also been reported to influence the autonomic response to hypoglycemia via opioid receptor activation [48]. The opioid receptor antagonist naloxone improved counterregulatory responses [49] and prevented HAAF [48] in humans. Therefore, both the inhibition of ACE activity and antagonism of opioid receptors may protect against angiogenesis and HAAF in humans, suggesting that angiogenesis, in which angiopoietin-1 is involved, could also be modulated by actions of opioids. Therefore, a possible explanation for the present results may be that genetic variations in the ANGPT1 gene could influence individual differences in angiogenic effects, which in turn may lead to individual differences in opioid analgesic efficacy.” [Line 386-398]

3) - Do we know anything on how these genetic variants influence the function of the receptor?

Response: Our GWAS identified several potent SNPs that were significantly associated with opioid requirements in cancer pain patients, including rs1283671 and rs1283720 in the ANGPT1 gene region, GSA-rs72752701 (rs72752701) in the additive and recessive models (Tables 2 and 4), and kgp11947181 (rs79524268), 9:133013640 (rs2417163), and rs763315292 in the dominant model (Table 3). Among these, the GSA-rs72752701, kgp11947181, and 9:133013640 SNPs are located in intergenic regions, whereas the rs763315292 SNP is located in the SEC23 interacting protein gene (SEC23IP) region. Although no significant eQTLs were found for the kgp11947181 and rs763315292 SNPs, the GSA-rs72752701 SNP is associated with mRNA expression of the heat shock protein family A (Hsp70) member 5 (HSPA5) gene, and the 9:133013640 SNP is associated with mRNA expression of the hemicentin 2 (HMCN2) and long intergenic non-protein coding RNA 963 (LINC00963) genes, according to the GTEx portal [36]. The SEC23IP, HMCN2, and LINC00963 genes have not been previously reported to be related to the opioid system or its receptors. In a previous quantitative mass spectrometry proteomics study of purified synaptosomes that were isolated from the caudate in two groups of rhesus macaques that were chronically infected with simian immunodeficiency virus (SIV) and differed with regard to morphine treatment regimens, the upregulation of HSPA5 was found in the SIV+morphine group, indicating an increase in cellular stress as a result of the SIV/morphine interaction, thereby leading to central nervous system dysfunction [50]. This previous study suggests that the expression and function of HSPA5 could interact with the opioid system, and alterations of HSPA5 mRNA expression through the GSA-rs72752701 SNP might influence the function of the opioid system, including opioid receptors. We added this discussion in the revised manuscript. [Lines 412-433]

4) - Please add two sentences more on how these results can be used in clinical practice or what other studies should be planned before we can think of any clinical application of these data.

Response: According to the reviewer’s suggestion, we added two sentences about how these results could be applied clinically and how future studies should be designed before clinical application. [Lines 471-478]
